# A 40-Year Cohort Study of Evolving Hypothalamic Dysfunction in Infants and Young Children (<3 years) with Optic Pathway Gliomas

**DOI:** 10.3390/cancers14030747

**Published:** 2022-01-31

**Authors:** Stefania Picariello, Manuela Cerbone, Felice D’Arco, Hoong-Wei Gan, Patricia O’Hare, Kristian Aquilina, Enrico Opocher, Darren Hargrave, Helen A. Spoudeas

**Affiliations:** 1Neuro-Oncology Unit, Department of Paediatric Oncology, Santobono-Pausilipon Children’s Hospital, 80123 Naples, Italy; s.picariello@santobonopausilipon.it; 2Department of Woman, Child and General and Specialized Surgery, University of Campania Luigi Vanvitelli, 80138 Naples, Italy; 3Department of Endocrinology, Great Ormond Street Hospital for Children, London WC1N 3JH, UK; HoongWei.Gan@gosh.nhs.uk (H.-W.G.); HelenS@gosh.nhs.uk (H.A.S.); 4Section of Molecular Basis of Rare Disease, University College London Great Ormond Street Hospital Institute of Child Health, London WC1N 1EH, UK; 5Department of Radiology, Great Ormond Street Hospital for Children, London WC1N 3JH, UK; 6Department of Oncology, Great Ormond Street Hospital for Children, London WC1N 3JH, UK; patricia.ohare@belfasttrust.hscni.net (P.O.); enrico.opocher@aopd.veneto.it (E.O.); Darren.Hargrave@gosh.nhs.uk (D.H.); 7Department of Neurosurgery, Great Ormond Street Hospital for Children, London WC1N 3JH, UK; Kristian.Aquilina@gosh.nhs.uk; 8Pediatric Hematology, Oncology and Stem Cell Transplant Division, Padua University Hospital, 35128 Padua, Italy

**Keywords:** optic pathway glioma, endocrine morbidity, infancy, pituitary, hypothalamus, diencephalic syndrome

## Abstract

**Simple Summary:**

Serious, poorly understood health issues affect young children with optic pathway tumours. We studied the risk of developing life-limiting hormonal, metabolic, and neurobehavioural disorders by tumour position, recurrence, and treatment, in those diagnosed under 3 years. We found the highest risk for future complex health issues in those presenting with failure to thrive, under one year of age, and/or a tumour involving a deep midbrain area called the hypothalamus. Time, repeated tumour growth, and salvage therapies (radiotherapy, surgery) contributed. We highlight the need for a better understanding of tumour-induced midbrain injury and for neurobehavioural and hormonal support to inform rehabilitation in the growing years, during and beyond cure, to optimise quality of life for these survivors and their families. This might inform oncology treatment strategies and determine new areas for support and collaborative neuroscience research in this high survival group.

**Abstract:**

Despite high survival, paediatric optic pathway hypothalamic gliomas are associated with significant morbidity and late mortality. Those youngest at presentation have the worst outcomes. We aimed to assess presenting disease, tumour location, and treatment factors implicated in the evolution of neuroendocrine, metabolic, and neurobehavioural morbidity in 90 infants/children diagnosed before their third birthday and followed-up for 9.5 years (range 0.5–25.0). A total of 52 (57.8%) patients experienced endo-metabolic dysfunction (EMD), the large majority (46) of whom had hypothalamic involvement (H+) and lower endocrine event-free survival (EEFS) rates. EMD was greatly increased by a diencephalic syndrome presentation (85.2% vs. 46%, *p* = 0.001)), H+ (OR 6.1 95% CI 1.7–21.7, *p* 0.005), radiotherapy (OR 16.2, 95% CI 1.7–158.6, *p* = 0.017) and surgery (OR 4.8 95% CI 1.3–17.2, *p* = 0.015), all associated with anterior pituitary disorders. Obesity occurred in 25% of cases and was clustered with the endocrinopathies. Neurobehavioural deficits occurred in over half (52) of the cohort and were associated with H+ (OR 2.5 95% C.I. 1.1–5.9, *p* = 0.043) and radiotherapy (OR 23.1 C.I. 2.9–182, *p* = 0.003). Very young children with OPHG carry a high risk of endo-metabolic and neurobehavioural comorbidities which deserve better understanding and timely/parallel support from diagnosis to improve outcomes. These evolve in complex, hierarchical patterns over time whose aetiology appears predominantly determined by injury from the hypothalamic tumour location alongside adjuvant treatment strategies.

## 1. Introduction

Paediatric low-grade gliomas (pLGGs) are the most common type of central nervous system (CNS) tumours in children [1,2,3]. They occur preferentially during the first decade of life, tend to stabilise after puberty, and develop in 15–20% of children with neurofibromatosis type 1 (NF1) [3]. Around 30–50% of pLGGs affect diencephalic structures—the optic nerves, chiasm, tracts, hypothalamus, and suprasellar midline—and are collectively termed optic pathway and hypothalamic gliomas (OPHGs) [4]. Their benign histology belies their often aggressive behaviour and recurrent treatments imposed in attempts to preserve vision, whilst any associated hypothalamic injury has not been possible to characterise alongside attempts to cure. High survival rates [4] mask parallel high morbidity rates which affect the quality of survival and late mortality. Visual impairment depends on tumour location, age at diagnosis, recurrence and hence repeated treatment rates, and NF1 status [5,6,7,8]. Additional neuroendocrine morbidity evolves over decades from likely early and increasingly manifest hypothalamic injury whose contribution to late mortality remains unclear [4,9,10,11,12,13,14,15]. The hypothalamic tumour location predicts the speed of onset of neuroendocrine deficits [9] and increases the incidence of anterior pituitary deficits and obesity [4]. Those youngest at presentation and those with hypothalamic involvement have the poorest long-term overall survival (OS), progression-free survival (PFS), and clinical morbidity [4,9,16]. However, there are few, if any, longitudinal studies of cohorts younger than 3 years old, in whom the classical hypothalamic diencephalic syndrome (DS) presentation is exclusively seen [17,18]. Thus, we aimed to study in detail the history of the evolution of hypothalamic quality of survival neuroendocrine, metabolic, and neurobehavioural outcomes in this very young population, against a background of increasing, repeated treatments cycles, to quantify the extent of—and ascertain which disease and treatment factors predicated—long-term morbidity outcomes that limit the quality of life.

## 2. Results

### 2.1. Demographic, Tumour Location, and Clinical Data at Presentation 

Our search identified 97 eligible children, but 7 who had moved away and had no endocrine follow-up data were excluded from the study. The remaining 90 were diagnosed with OPHG at a median inter-quartile range (IQR) age of 1.84 (0.83–2.51) (0.06–3) years and followed for 9.5 (4–12.5) (0.5–25.0) years until aged 10.8 (5.7–14.2) (0.6–26.3) years; one-third (30: 33.3%) were under 1 year of age at diagnosis, of whom almost all (26/30, 86.7%) had post-chiasmatic and hypothalamic (28/30, 93.3%) involvement; the last feature was uniquely associated with DS (27/90: 30%) and raised intracranial pressure (RICP) (22/90: 24%). Of the total cohort, 42/90 (46.7%) were male, with a preponderance of modified Dodge classification (MDC) 3/4 tumours (46: 51.1%) and hypothalamic involvement (H+) (56: 62.2%). Just 19 (21.1%) had MDC1 and 25 (27.8%) MDC2 tumours, and 11 (12.2%) presented with leptomeningeal metastasis (LM+).

Approximately a third (27/90: 30%) had a clinical and/or genetic diagnosis of NF1, of whom all but one presented after the age of 1 year (median age at presentation 1.34 (1.94 –2.85) (range 0.91–3.00) years), and predominantly with anterior tumours (12 MDC1 and 11 MDC2 with H+ in 3 cases), few having post-chiasmatic involvement (four MDC3/4 with H+ in one case). Patients’ characteristics and presenting features are shown in Table 1.

### 2.2. Treatment Burden and Survival Outcome

Most patients (66/90: 73.3%) underwent first-line treatment with chemotherapy according to International Society for Paediatric Oncology (SIOP) protocols in use at the time.

Six patients diagnosed before 1990 (6.7%) received first-line focal cranial radiotherapy (RT). A third (31: 34.4%) required surgical debulking (15), biopsy (11), 13/26 with additional decompressive procedures, or decompression alone (5). A minority (14: 15.5%) with anterior pathway tumours were placed on a surveillance strategy.

Pathology was available in 48 children who underwent surgery: 39 tumours were pilocytic astrocytoma, 5 pilomyxoid astrocytomas, 3 gangliogliomas, and 1 fibrillary astrocytoma.

Over half progressed (51: 56.7%) at a median of 1.9 (1.3–3.2) (0.1–11.13) years from diagnosis; of these, 41 relapsed between 2 and 7 more times, whilst 8 (10%) died of their disease or acute/chronic complications, RICP (1), or CSWS (1), 3.9 (2.5–16.1) (0.52–21.74) years later, aged 4.96 (2.76–18.03) (0.58–23.66) years. Specific causes of death were acute intratumoural haemorrhage = 1, cerebrovascular complication = 1, tumour progression = 4, and cerebral oedema and brainstem herniation = 2.

In addition, 5- and 10-year OS rates were high at 93.7% and 92%, respectively, but declined up to 21.74 years from diagnosis, similar to the respective 5- and 10-year PFS rates of 42% and 40.5%, which declined to 36.8% at 11.1 years after diagnosis (Figure 1A). 

Chemotherapy (between one (28.9%), two (10%), and up to six (1.1%) lines of treatment) was the commonest treatment (68: 75.5%), whereas 21 (23.3%) eventually received RT at a median age of 5 (3.1–8.36) 0.7–1.36) years; 16 received both chemotherapy and RT, but 12 (13.3%) subjects remained treatment naïve throughout a median of 6.6 (3.8–10.8) (0.7–14.6) years of follow-up.

### 2.3. Endo-Metabolic Dysfunction 

In this study, 52 out of 90 children (57.8%) had some degree of EMD. Median time to the first endocrine event was 3.4 (1–6) (0.05–13.64) years, with 5- and 10-year endocrine event-free survival (EEFS) rates of 54.1% and 31.4%, respectively. EEFS continued to decline up to 13.6 years after diagnosis (Figure 1A).

Of 52 subjects experiencing EMD, 46 (88.5%) had H+ and lower event-free survival (EFS) for anterior pituitary disorder (APD), posterior pituitary disorder (PPD), and metabolic disorder (MD) (Figure 2A–C).

Only 10/56 cases with H+ at diagnosis had no evident EMD, but their follow-up was shorter, at 3.6 (1.6–5.8) (0.5–10.8) years of follow-up.

Both NF-1 and anterior tumour position were more common in those 38/90 (42.2%) without EMD, of whom half (50%) had NF-1, 19 (21.1%) and 9 (10%) had MDC1 and MDC2, respectively, whilst only a minority (10: 11.1%) had MDC3/4. 

Out of 90 (52.2%) subjects, 47 developed at least one APD (median (range) 2(1–5)) in conjunction with MD in 21 (23.3%) and PPD in 12 (13.3%).

### 2.4. The Effect of Diencephalic Syndrome on Endo-Metabolic Disorders

The diencephalic syndrome was present in a third (27/90: 30%) of children at diagnosis and almost exclusively in non-NF1 patients, with the exception of just one case who had NF-1. They were significantly younger than those without DS and more likely to be under 1 year of age (18/27 (66.7%) vs. 12/19 (19%) *p* < 0.0001). All had hypothalamic involvement (H+) and a more posterior tumour position—MDC3/4 (85.2% vs. 36.5%, *p* < 0.0001) and LM+ (33.3% vs. 3.2%, *p* < 0.0001) (Appendix A). Except for the presence of DS, there were no other endo-metabolic disorders noted at presentation, but these patients had the highest evolution to EMD at last follow-up, compared with those without DS (23/27 (85.2%) vs. 29/63 (46%), *p* = 0.001) (Appendix A)), with earlier onset of growth hormone deficiency (GHD) (hazard ratio (HR) 2.5 (95% confidence interval (CI) 1.3–4.8), *p* 0.007), thyroid stimulating hormone deficiency (TSHD) (HR 2.98 (95% CI 1.26–7.07), *p* 0.013), central precocious puberty (CPP) (HR 5.4 (95% CI 2.4–12.3), *p* < 0.001), and obesity (HR 2.34 (95% CI 1.03–5.36), *p* 0.043) (Appendix A). 

### 2.5. Anterior Pituitary Disorders (APDs)

Over half (48/90) of children developed APD at 3.9 (1.7–6) (0.05–10.7) years after tumour diagnosis, and 5- and 10-year APD-EFS were 57.3% and 33.7%, respectively (Figure 1B). GHD was the most common APD (41.1%), while ACTH was much less common, allowing for the late evolution of Gn deficiency only manifest by age (Figure 3A, Table 2). 

Of 36 children with GHD (66.7%), 24 received GH replacement therapy for a median (range) of 4 (0.5–20.8) years at physiological doses of between 19.5 (6–37) µg/kg/day and 25.15 (12.6–39.0) µg/kg/day, with no differences in number of children whose disease progressed or in number of progressions per patient between treated (21/24 = 87.5%, median 3.0, range 0.0–7.0) and untreated (11/12 = 91.7%, median 3.0, range 1.0–5.0) GHD patients.

CPP was the earliest endocrine dysfunction and second in prevalence to GHD. It affected 26.7% of our cohort overall (10 females and 14 males, a median of 3.8 years after OPHG diagnosis, with the CPP-specific EFS curve declining to 65.1% at 7.8 years of follow-up) (Figure 3A, Table 2). All patients with CPP were treated with GnRH analogues at a median age of 5.7 (4.7–6.6) (2.7–8.5) years in females and 4.8 (2.7–6.6) (2.1–8.9) years in males.

GnD was, as yet, the least common APD. It occurred in 6 females (6.7%) and 11 males (15.5%) at 14 (13.5–14.1) (13.4–14.1) years and 14.2 (14.0–14.5) (14.0–15.2) years of age, respectively. However, there were only 13 females older than 13 years and 13 males older than 14 years at the last evaluation in our cohort in whom this diagnosis could be potentially made. The GnD-specific EFS curve only began to decline 9.4 years after diagnosis, reaching 22.2% at 14 years of follow-up. In 8 cases out of 17 (47%), GnD evolved from a previous CPP.

TSHD occurred in 21 subjects (23.3%) and at a similar prevalence to ACTHD, which occurred in 19 (22%) patients. Both ACTHD and TSHD survival curves continued to decline up to 14.7 years after diagnosis (Figure 3A, Table 2).

Age <1 year at diagnosis, DS, hydrocephalus, absence of NF1, MDC3/4, and H+ were independent predictors at OPHG diagnosis of future development of APD in univariate Cox regression analysis. After adjusting for the same factors, only H+ remained significant in multivariate analysis (Table 3). 

With increasing time, the number of progressions, chemotherapy, radiotherapy, surgery, and follow-up time itself predicted long-term APD, but radiotherapy and surgery prevailed on the other covariates in multivariate analysis (Table 3).

We thus explored the role of significant factors in the above Cox and logistic regression multivariate analyses (radiotherapy and surgery against hypothalamic involvement) and found a highly significant effect of hypothalamic involvement (OR 6.1, 95% CI 1.7–21.7, *p* 0.005), as well as radiation (OR 16.2, 95%CI 1.7–158.6, *p* 0.017) and surgical intervention (OR 4.8 95% CI 1.3–17.2, *p* 0.015), to influence the onset of APD, with no factor prevailing on the others. 

### 2.6. Posterior Pituitary Disorders (PPDs)

Posterior pituitary disorders occurred in 15 subjects (16.7%), only after therapeutic interventions and/or as a consequence of hydrocephalus in those with suprasellar tumours (MDC2/3) and hypothalamic disease. Here, 5- and 10-year PPD-EFS were 87.4% and 80.6% (Figure 3B, Table 2).

In one child, syndrome of inappropriate antidiuretic hormone secretion (SIADH) presented as a complication of vincristine chemotherapy, in seven children, this followed debulking or decompressive surgery—in three of whom it was in the context of a triphasic response—and in two subjects, it was attributed to tumour progression causing RICP and subsequent death.

Central diabetes insipidus (CDI) requiring desmopressin (DDAVP) occurred in eight children (8.9%) and was seen exclusively after surgery (resection of any extent, biopsy, or surgery for hydrocephalus), and in the context of a triphasic response in three out of eight. Six out of eight children went on to require permanent DDAVP replacement.

Severe central salt wasting syndrome (CSWS) was also only observed in one child in the context of a post-operative triphasic response and imminent death.

Six out of nine children who died had a history of transient or permanent PPD.

### 2.7. Metabolic Disorders (MDs)

MDs were diagnosed in 23 cases (25.5%) with an MD-EFS of 93.6% and 70% at 5 and 10 years, respectively, with a survival curve declining up to 44.8% at 15 years (Figure 3C, Table 2). 

In this context, 23 children (25.5%) became obese after a median 6.8 (5.2–9.5) (0.7–14.7) years from diagnosis, almost half (11/23) of whom then suffered frank glucose dysregulation manifest as impaired glucose tolerance (IGT) (10) or type2 diabetes mellitus (T2DM) (1) at a median 3.8 (1.0–5.2) (0–14.2) years later. 

Furthermore, all but two obese children suffered additional APD disorders and had a significantly higher endocrine morbidity score (median M-EMS 5.0, IQR 3.0–6.0, range 1.0–7.0) than those with normal BMI (median 0.0, IQR 0.0–1.0 range 0.0–5.0), *p* < 0.0001.

Predictors of time to MD in univariate Cox regression analysis were MDC stage, hypothalamic involvement, DS, and hydrocephalus (Table 4).

Factors and covariates associated with MD in long-term follow-up were radiotherapy, number of concomitant APD, and follow-up duration. The number of concomitant APD remained the only significant variable in multivariate binary logistic regression analysis (Table 4).

### 2.8. Neurobehavioural Morbidity

The cerebrovascular disease occurred in nine (10%) subjects—three of whom developed moyamoya disease attributed to radiotherapy and six suffered a stroke (four after radiotherapy and two after surgery). Motor disorders were present in 17 (18.9%), of whom 6 suffered hemiparesis, 2 paraplegia, and 9 residual weakness. Epilepsy requiring anti-epileptic medication for a seizure disorder and/or abnormal electroencephalogram occurred in 20 (22.2%), Neurobehavioural deficits occurred in 52/90 (57.8%) children. Attention and behavioural problems occurred in 24 (26.7%) and 27 (30%) children, respectively.

Among 70 school-aged children at the latest follow-up, 42 (46.7% of the whole population) attended mainstream education, but 14 (15.6%) of these required additional funded schooling support, whilst a further 28 (31.1%) were eventually placed in a special-needs school to meet their educational need.

Metabolic dysfunction (EMD) was significantly more prevalent in those with neurobehavioural complications (39/52: 75%) than in those without (12/38: 31.6%) (*p* < 0.0001), coupled with a higher neuroendocrine morbidity score (M-EMS; median (range) 2.5 (0–7) vs. 0 (0–5), *p* < 0.0001). Both MDC stage and H+ increased the risk of neurobehavioural disorders (OR 2.2, 95% C.I. 1.2–3.8, *p* = 0.007 and OR 2.5, 95% C.I. 1.1–5.9, *p* = 0.043, respectively).

Radiotherapy was the sole treatment modality associated with neurobehavioural morbidity (OR 23.1, 95% C.I. 2.9–182, *p* = 0.003). Other covariates associated with increased odds of neurobehavioural morbidity were the total number of progressions (1.3, 95% CI 1.4–1.8, *p* 0.024) and the length of follow-up in years (1.3, 95% CI 1.2–1.5, *p* < 0.0001).

## 3. Discussion

Infants with OPHG are recognised to have a more difficult and unpredictable disease course than older children, characterised by higher rates of disease progressions, poorer visual outcomes, and complex chronic neuroendocrine morbidity that jeopardises the quality of their otherwise high survival [5,19,20,21,22]. However, the aetiology of the neuroendocrine dysfunction, its longitudinal evolution over decades from infancy to maturity, and its contribution to long-term mortality and morbidity have not been systematically studied in international trials of oncology treatment, as they are largely attributed to the detrimental effects of radiation on the young brain [4,15,23]. This is particularly relevant to tumours involving the vital, primitive deep diencephalic hormone signalling pathways, which especially affect infants, and a proportion present with DS [18].

In this longitudinal study of 90 infants and young children diagnosed with OPHG before their third birthday and followed for a median of 9.5 but up to 25 years, we report the long-term neuroendocrine and metabolic morbidity of this very young population as it evolves with developmental age, tumour location, and disease course. We comprehensively investigated the onset of anterior and posterior pituitary dysfunction as well as metabolic disorders and documented clinically important neurobehavioural dysfunction. By detailing neuroendocrine outcomes, we provide an additional morbidity marker and evidence for primary, potentially devastating tumour-induced hypothalamic brain injury, aggravated by disease progression and treatment failure.

Disease progression affected more than half of our cohort and 10% died a median of 3.2 years later, in agreement with previous reports [20,21,22].

Most tumours in this young cohort were post-chiasmatic, as previously reported [5,24], but nevertheless, the prevalence of hypothalamic involvement was particularly high (>60%), and almost invariably present in those under 1 year of age. Such a sizeable infant hypothalamic cohort emphasises the importance of early and progressive tumour-induced hypothalamic injury to the later hierarchical evolution of downstream pituitary deficits over time and to the poorer clinical and neurobehavioural outcomes seen in these children. 

The median time from diagnosis to the first endocrine event (usually GHD or CPP) was over 3 years in this cohort but continued to decline up to 13.6 years, whilst the hierarchical evolution of anterior pituitary deficits we previously reported, in which GHD is most sensitive and ACTH most robust [9] were again evident, allowing for the predictably very low prevalence of GnD in this group of still average prepubertal age at follow-up (10.8 years). This evolution of endocrinopathy with time after tumour injury is only evident with longitudinal clinical follow-up which includes a consistent endocrine strategy and long-term observation from infancy to maturity. 

In order to analyse the effect of presenting DS on our cohort’s outcomes, we were obliged to artificially exclude DS from our definition of endo-metabolic disorders. Thus, no patient was deemed to present with neuroendocrine dysfunction other than DS, but CPP also occurred notably early. This finding could be genuine given the young age of our cohort. Both DS and CPP are recognised early disorders of the hypothalamic hunger/satiety and gonadostat control mechanisms, respectively. However, the lack of systematic baseline and dynamic endocrine testing in this vulnerable population, often presenting acutely unwell, with prioritisation of oncological treatments, could also contribute to this result, highlighting the importance of a neuroendocrine specialist review for all these complex cases, not yet routine. 

Infants with DS are also recognised to present with GH profiles suggestive of GH excess [25], which also inevitably spontaneously evolves to GHD, a highly prevalent anterior pituitary deficiency in this and other cohorts. 

It is highly possible that DS itself represents the early infant phase of a biphasic hypothalamic disorder affecting GH and hunger/satiety signalling pathways, later leading to the high prevalence of GHD and obesity which are only evident with very long longitudinal follow-up throughout the age span and beyond adulthood. 

This is also the most likely explanation for the apparently delayed onset of GHD in this cohort contrary to other reports [9,26]. Given the first 1–2 years of growth are dependent on nutrition and growth factors, not GH secretion, isolated GH deficiency is not usually diagnosed in infancy. Moreover, high disease progression rates are likely to have downgraded both the prioritisation of growth assessment and GH substitution to those found deficient. In fact, we documented GHD in most (60%) of our DS cohort, strongly suggesting APD evolves from a primary tumour-induced hypothalamic injury incurred at diagnosis, evolving to affect the downstream pituitary reserve with time and increasing tumour burden, to which salvage therapies contribute. 

Nevertheless, we again found no difference in disease progression rates between GH-treated and untreated groups in this vulnerable cohort, despite a proactive GH treatment policy at our institution. This reinforces the concept that GH replacement does not need to be unnecessarily delayed due to concerns about tumour progression, provided only timely replacement doses are used in a proven deficient child [27,28,29,30]. Current guidelines [31] suggest a period of tumour stability (at least one year) before introducing GH treatment. However, given the relapsing–remitting nature of the disease in young children, the metabolic implications of the lack of anabolic GH in the growing child, and the need to positively improve health-related quality of survival in this particular group, earlier GH substitution requires consideration.

In this young cohort, it also became clear that the earliest endocrine disturbance was CPP with an atypical preponderance of males and second in prevalence to GHD. We again confirmed our previous report [9] of an unexpected biphasic evolution of CPP to later Gn deficiency in the same patients, but a greater proportion of this cohort was affected (47% vs. 37%) despite having younger age (10.5 years) at follow-up. We postulate an evolving hypothalamic dysregulation to explain this previously unrecognised biphasic phenomenon: any brain injury may disrupt the hypothalamic ‘brake’ and GnRH pulse generator to cause CPP in a prepubertal population, but wider disease and further injury due to progressions and further local treatments to this area may cause Gn deficiency in a peri- or post-pubertal cohort. 

The prevalence of TSH deficiency and, importantly, of life-threatening ACTH/cortisol deficiency signifying additional severe panhypopituitarism were again least prevalent but slightly higher in this cohort, compared with previous reports [9,32]. We attribute this to the significantly higher prevalence of hypothalamic involvement in this exclusively very young cohort (62% vs. 40%), suggesting these are less well-recognised endo-metabolic evolutionary characteristics of an acquired hypothalamic injury occurring at different developmental windows of time.

Despite the high prevalence of hypothalamic involvement, posterior pituitary dysfunction rates were again comparatively low, apparent only after therapeutic interventions, especially any type of surgery (including biopsy and shunts), where there was already pre-existing hypothalamic disease. They were mainly transitory, sometimes as part of a triphasic perioperative course, whilst permanent CDI was rare, consequent upon surgery, and similar to our older series of patients [9]. Despite their rarity, PPD disorders were relatively prevalent in children who ultimately died and/or one of the factors contributing to death in some severe cases. 

Obesity was clustered with concomitant endocrine disorders in almost all cases and affected 25% of our population. It took a median of 7 years from OPHG diagnosis for patients to develop obesity, reflecting the young age at onset in our cohort. A third (30%) evolved to obesity from a cachectic DS baseline in another biphasic hypothalamic evolutionary phenomenon. The obesity survival curve for DS patients progressively declined from diagnosis to cross that of non-DS 5 years later, continuing to drop thereafter so that the DS group was significantly more obese at 10 years. Overall, the MD survival curve echoed that of APD but with a 3–5 year time delay. 

Presenting DS carried a much higher risk for future endo-metabolic disorders (GHD, TSHD, CPP, and obesity), suggesting children with either DS and/or hypothalamic involvement should be prioritised for closer neuroendocrine follow-up from diagnosis, with timely diagnosis and treatment of any endocrine dysfunction to maintain growth, lean-to-fat-mass ratio, aid recovery, and well-being, as well as to maintain the quality of life, despite ongoing oncology treatment.

We found better neuroendocrine outcomes in children with NF-1-related OPHG. A diagnosis of NF-1-related OPHG is rare before 3 years of age [26,33]. In our young cohort, its presence was associated with more MDC1 disease (50%) and few (15%) hypothalamic tumours, in line with previous reports [26,33], which most likely accounts for these neuroendocrine outcomes.

Taken together, and allowing for treatment selection bias and salvage therapies in our young cohort, our results indicate that hypothalamic tumour location is most probably the strongest predictor of both neuroendocrine and metabolic dysfunction, evolving from hypothalamic dysregulation in a biphasic pattern to later pituitary deficits and likely to further increase with time. The independent contribution of radiotherapy and surgery, often imposed as salvage therapies in worsening disease, are difficult to separate and subject to disease bias/confounder effects but contribute to APD and evolving hypothalamic injury over time. Radiation additionally affects metabolic dysfunction, whilst surgical intervention and/or hydrocephalus precipitated PPD only in those with hypothalamic involvement. The impact of tumour stability or shrinkage on any evolving neuroendocrine dysfunction is important to ascertain but difficult to achieve long-term, especially when individualised sequential treatment interventions are applied to an age-dependent biphasic evolutionary disease course. This would require future trials designed with this important outcome in mind, targeting this specific, high-risk age cohort for early neuroendocrine and neurodevelopmental support, followed carefully throughout the age span. 

Poorer neurological deficits and psychological disorders clustered with endo-metabolic conditions, suggesting a common hypothalamic injury denominator from tumour site and disease progression before local salvage treatments were imposed. 

One-third of our cohort experienced inattention/behavioural disorders, whilst over half required special needs educational support of varying degrees, and a further fifth needed supported education in the mainstream. These sobering figures may yet represent an underestimate, given the absence of systematic assessments in children undergoing repeated therapy and the tendency for neurobehavioural disorders to become more evident with puberty, time, and curricular demand. 

Our study was retrospective in a highly selected cohort of young children chosen specifically to determine the aetiology of their disproportionate morbidity, undergoing heterogeneous treatments and inconsistent neuropsychological assessments over the decades. Nevertheless, though it still lacks further maturity, it represents the largest and longest longitudinal study of DS and/or very young children with hypothalamic gliomas correlated with neuroendocrine and neurodevelopmental outcomes over time. It has, for the first time, highlighted the emerging biphasic nature of injuries to hypothalamic hormone signalling incurred by the time of tumour diagnosis and evolving to pituitary deficits with time, systemic treatment failures, and disease progression, as well as local treatment burden. This hypothalamic and biphasic brain injury is greatest in diencephalic infants, evolving slowly and inexorably despite treatment and resulting in a significantly reduced quality of life despite prolonged survival [34,35,36]. Further comorbidities whose time-line is necessarily a later consequence (GnD, obesity/glucose dysregulation, neurobehavioural morbidity) are most likely to emerge in the future and emphasise the importance of very long-term follow-up data from infancy to beyond adult maturation, before outcomes and their aetiology can be accurately defined. Such comparator datasets will also provide information on the impact of novel targeted therapies (BRAF and MEK inhibitors) on similar tumour control, neuroendocrine, and neurobehavioural outcomes in this vulnerable group and require intensive collaborative efforts.

## 4. Materials and Methods

This was a single-centre, retrospective longitudinal cohort study of patients diagnosed with OPHG before their third birthday, identified from clinical electronic databases at Great Ormond Street Hospital, London, UK, and followed over 4 decades of changing treatment strategies, between January 1981 and December 2020.

Patients were searched by diagnosis in our electronic database using the following search criteria: ‘optic pathway glioma’, ‘optic nerve glioma’, ‘hypothalamic glioma’, ‘optic glioma’, and ‘chiasmatic glioma’. 

The list of patients identified was further matched against a clinical database held by the responsible endocrine clinician for these patients.

Only those with pathological and/or neuroradiological confirmation were included.

Demographic and relevant clinical data (presenting symptoms, treatment type and number, tumour recurrence/progression, endocrine, metabolic and neurocognitive dysfunction) were recorded.

Magnetic resonance images (MRIs) were reviewed by an experienced neuroradiologist (F.D.A.), and tumour location was assessed according to the modified Dodge classification (MDC) [37].

Tumours confined to the optic nerve were classified as MDC1, and those involving the optic chiasm as MDC2, whereas post-chiasmatic involvement was recorded as MDC3/4 (optic tracts and/or optic radiation). Tumours involving multiple regions were assigned the highest (most posterior) MDC stage, in agreement with previous studies [26]. Hypothalamic involvement (H+) and leptomeningeal metastases (LM+) were independently recorded.

Diencephalic syndrome (DS) was defined as failure to thrive not explained by other causes and the downward crossing of 2 centile bands for weight with a normal growth rate, or body mass index (BMI) ≤2 standard deviations (SD) with/without emaciation, euphoria, hyperactivity, and in the presence of confirmed H+ [25,38,39]. Characteristics and outcomes of children presenting with DS were compared with those without DS.

Raised intracranial pressure (RICP) was defined by the presence of clinical signs and symptoms of intracranial hypertension together with MRI findings suggestive of the same.

All patients underwent clinical Tanner staging, auxological measurement, baseline, and dynamic assessment of pituitary function at the first signs of growth deceleration from baseline, or when growing but at the expense of increased weight gain, and/or at the time of the first onset of puberty. At the onset of obesity (BMI > +2SDS), patients were further assessed by a basal and 2 h insulin and glucose response to a glucose load (OGTT).

To assess the degree of endocrine dysfunction and associated metabolic comorbidities at the latest follow-up, we used the Modified Endocrine Morbidity Score (M-EMS) from DeVile et al. [40] and Gan et al. [9], consisting of a potential total of 10 hypothalamic–pituitary and metabolic dysregulations: growth hormone deficiency (GHD), central precocious puberty (CPP), gonadotropin (LH/FSH) deficiency (GnD), TSH deficiency (TSHD), ACTH deficiency (ACTHD), central diabetes insipidus (CDI), syndrome of inappropriate anti-diuretic hormone (SIADH) secretion, cerebral salt wasting syndrome (CSWS), obesity (BMI > +2SDS), impaired glucose tolerance (IGT), or type 2 diabetes mellitus (T2DM).

Neurological deficits, the presence of epilepsy, and cerebrovascular disease (clinical and radiological diagnosis of ischaemic or haemorrhagic stroke and/or moyamoya disease) were also collected. The requirement for supported or special educational needs schooling, together with the diagnosis of attention and behavioural disorders, were used as surrogates for neurobehavioural outcomes.

Treatment modalities were chemotherapy, radiotherapy, and surgery, including resection, biopsy, and decompressive procedures (cyst drainage, ventriculoperitoneal shunt, endoscopic third ventriculostomy, Ommaya Reservoir) [41].

### Statistical Analysis

Categorical variables are reported as numbers (percentage). Continuous non-parametric variables are presented as median (interquartile range) (range). Differences between groups were compared by chi-squared or Fisher’s exact tests for categorical variables and the Mann–Whitney U test for continuous non-parametric variables. Kaplan–Meier survival curves were calculated for progression-free survival (PFS), overall survival (OS), and endocrine event-free survival (EEFS), as well as event-free survival for each hypothalamic–pituitary and metabolic disorder. Progression was defined as any significant neuroradiological and/or clinical progression requiring treatment initiation or modification. EEFS was defined as the time to first neuroendocrine/metabolic event after diagnosis. Patients without events were censored at last follow-up or death for all outcomes. The log-rank test was used to compare survival curves. Univariate and multivariate Cox regression analyses were run to explore predictors of future endo-metabolic disorders, present at diagnosis.

Logistic regression was used to explore the impact of pre-determined factors and covariates occurring during follow-up on anterior pituitary dysfunction (APD), metabolic dysfunction (MD), and learning and neurological outcomes: radiotherapy, chemotherapy, surgery, years of follow-up, and the number of progressions. In the MD logistic regression analysis, the number of concomitant APD was also included. 

For both logistic and Cox regression analyses, variables that were statistically significant in the univariate analyses were included in the multivariate analyses.

Hazard ratio (HR) and odds ratio (OR) are presented with 95% confidence interval (CI).

Statistical analyses were conducted using SPSS 22 IBM Software.

## 5. Conclusions

Children presenting with hypothalamic involvement and OPHG before their third birthday carry a high and inexorable tumour-induced risk for future endocrine and metabolic morbidity, which evolves in a biphasic phenomenon from early hypothalamic dysregulation to later pituitary deficits and from infancy to maturity, evident only in very long-term data collection. Those presenting with DS and under 1 year of age are particularly vulnerable, whilst disease progression and local salvage therapy contribute. High frequencies of neurobehavioural disturbances cluster with endo-metabolic disturbance, suggesting a common hypothalamic aetiology from brain injury to this area which remains poorly understood and unsupported and needs further collaborative endo-oncology and neuroscience research. Overall survival is high (90%), emphasising the importance of reducing morbidity, but continues to decline up to 21 years from diagnosis, echoing early PFS rates and evolving life-threatening endo-metabolic morbidity, whose contribution to mortality remains unquantified. Early endocrine and neurodevelopmental referral with parallel, timely and ongoing endocrine replacement, and complex health assessments are necessary to inform education and health care planning in this growing and vulnerable population, during and beyond cure, to reduce long-term morbidity and to optimise quality of life for survivors and their families. 

## Figures and Tables

**Figure 1 cancers-14-00747-f001:**
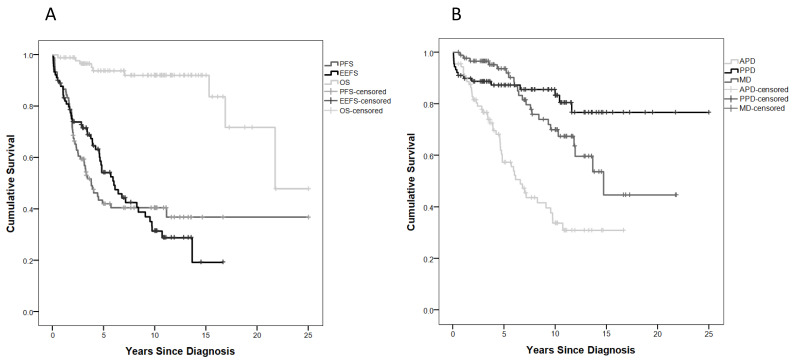
(**A**) Progression-free survival (PFS), overall survival (OS), and endocrine event-free survival (EEFS) curves; (**B**) event-free survival (EFS) curves for anterior pituitary disorder (APD), posterior pituitary disorder (PPD), and metabolic disorder (MD).

**Figure 2 cancers-14-00747-f002:**
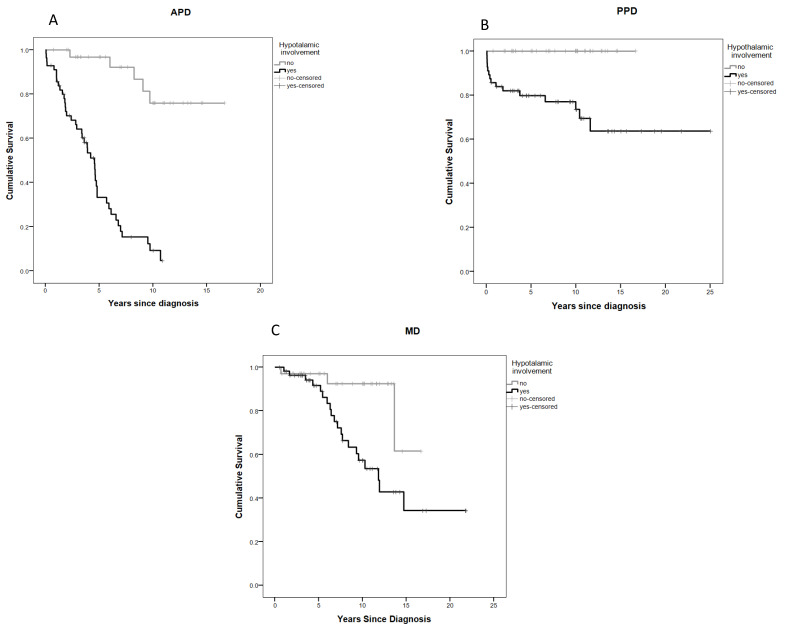
Event-free survival (EFS) curves: (**A**) anterior pituitary disorder (APD) by hypothalamic involvement; (**B**) posterior pituitary disorder (PPD) by hypothalamic involvement; (**C**) metabolic disorder (MD) by hypothalamic involvement.

**Figure 3 cancers-14-00747-f003:**
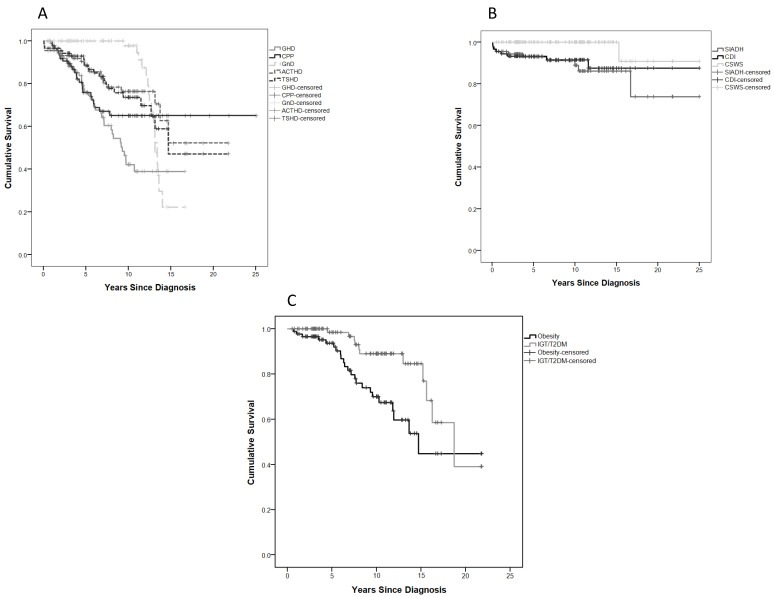
EFS survival curves: (**A**) specific anterior pituitary disorder (APD); (**B**) posterior pituitary disorder (PPD); (**C**) metabolic disorder (MD).

**Table 1 cancers-14-00747-t001:** Demographic characteristics and presenting features of our cohort.

	Study Population (*n* = 90)
Male	42 (46.7)
Age at diagnosis (years)	1.84 (0.83–2.51) (0.06–3.00)
≤1 year	30 (33.3)
NF1 positive	27 (30.0)
Symptoms at diagnosis	
Visual symptoms	41 (45.6)
Diencephalic syndrome	27 (30.0)
RICP	22 (24.4)
Asymptomatic	14 (15.5)
Radiological Classification	
MDC 1	19 (21.1)
MDC 2	25 (27.8)
MDC 3/4	46 (51.1)
Hypothalamic involvement	56 (62.2)
Leptomeningeal metastases	11 (12.2)
Follow-up (years)	9.49 (4.00–12.53) (0.52–25.00)
Deceased	9 (10)
Death age (years)	5.6 (2.76–18.03) (0.58–23.66)

Categorical variables are reported as number (percentage). Continuous nonparametric variables are presented as median (interquartile range) (range). NF1: neurofibromatosis type 1; RICP: raised intracranial pressure; MDC: modified Dodge classification.

**Table 2 cancers-14-00747-t002:** Endocrine free-survival rates (EFS) and median age at onset of anterior pituitary disorder (APD), posterior pituitary disorder (PPD), and metabolic disorder (MD).

	Number of Cases	5-Year EFS	10-Year EFS	Time Since Diagnosis (Years)	Age at Diagnosis (Years)
APD					
GHD	37 (41.1)	76.1	42.2	5.3 (2.8–8) (0.79–10.7)	7 (4.9–9.2) (1.5–12.5)
CPP	24 (26.7)	75.8	65.1	3.8 (1.9–5.3) (1.02–7.8)	5.6 (3–6.5) (2.1–8.9)
GnD	17 (18.9)	100	97.8	12.7 (11.8–13.3) (9.4–14)	14 (14–14.3) (13.4–15.2)
ACTHD	19 (22.2)	88.3	76.3	5.3 (1.7–7.4) (0.05–14.7)	7 (3.5–9.2) (0.3–15)
TSHD	21 (23.3)	88.8	76.4	5.9 (2.7–8.9) (0.05–14.7)	7.8 (4.9–10.8) (0.3–15)
PPD					
SIADH	10 (11.1)	92.9	88.9	2.8 (0.2–10.1) (0–16.7)	3.9 (1.1–11.3) (0.8–18.8)
CDI	8 (8.9)	93.3	91.5	0.8 (0.1–5.4) (0.05–11.6)	2.9 (2.3–7.6) (0.3–13.46)
CSWS	1 (1.1)	100	100	15.28	17.07
MD					
Obesity	23 (25.5)	93.6	70	6.8 (5.2–9.5) (0.7–14.7)	8.9 (6.9–10.8) (2.8–16)
IGT/T2D	11 (12.2)	98.4	89	8.1 (7.5–15.6) (4.5–18.7)	11.1 (9.2–17) (6.8–19)

Categorical variables are reported as number and percentage. Continuous nonparametric variables are presented as median (interquartile range) (range). Growth hormone deficiency (GHD), central precocious puberty (CPP), gonadotropin (LH/FSH) deficiency (GnD), ACTH deficiency (ACTHD), TSH deficiency (TSHD), central diabetes insipidus (CDI), syndrome of inappropriate anti-diuretic hormone (SIADH) secretion, cerebral salt wasting syndrome (CSWS), obesity (BMI> +2SDS), impaired glucose tolerance (IGT), or type 2 diabetes mellitus (T2DM).

**Table 3 cancers-14-00747-t003:** Predictors, factors, and covariates associated with APD. Cox regression results are reported as HR (95% CI). Binary logistic regression results are reported as OR (95% CI).

	Univariate		Multivariate	
	**HR (95% CI)**	** *p* **	**HR (95% CI)**	** *p* **
Absence of NF1	5.3 (2.2–12.5)	<0.0001	1.5 (0.5–4.4)	0.414
Age ≤ 1 year	2.3 (1.3–4.2)	0.007	0.7 (0.3–1.4)	0.317
Diencephalic syndrome	3.2 (1.8– 5.9)	<0.0001	1.6 (0.8–3.2)	0.137
Hypothalamic involvement	11.7 (4.5–30.2)	<0.0001	4.2 (1.3–13.8)	0.018
MDC	3.1 (1.9–5.1)	<0.0001	1.9 (0.9–3.7)	0.065
Hydrocephalus	2.7 (1.4–4.9)	0.002	1.1 (0.6–2.1)	0.775
	**OR (95% CI)**	** *p* **	**OR (95% CI)**	** *p* **
Radiotherapy	31.1 (3.9–245.5)	0.001	15.7 (1.2–211.8)	0.038
Chemotherapy	4 (1.4–11.6)	0.009	5.3 (0.9–28.7)	0.057
Surgery	12.8 (4.4–36.9)	<0.0001	9.9 (2.6–37.3)	0.001
Number of progressions	1.9 (1.4–2.6)	<0.0001	1.1 (0.8–1.6)	0.567
Years of follow-up	1.2 (1.1–1.3)	0.003	2 (1.1–3.8)	0.034

APD: anterior pituitary disorder; OPHG: optic pathway hypothalamic glioma; MDC: modified Dodge classification, HR: hazard ratio, CI: confidence interval, OR: odds ratio.

**Table 4 cancers-14-00747-t004:** Predictors, factors, and covariates associated with MD. Cox regression results are reported as HR (95% CI). Binary Logistic regression results are reported as OR (95% CI).

	Univariate		Multivariate	
	**HR (95% CI)**	** *p* **	**HR (95% CI)**	** *p* **
Absence of NF1	2.1 (0.7–6.3)	0.169	NI	NI
Age ≤ 1 year	1 (0.4–2.4)	0.989	NI	NI
Diencephalic Syndrome	2.3 (1.1–5.4)	0.043	1.2 (0.5–3)	0.679
Hypothalamic involvement	4.4 (1.3–14.9)	0.017	2.1 (0.4–10.2)	0.359
MDC	2.2 (1.1–4.4)	0.024	1.2 (0.5–3.1)	0.623
Hydrocephalus	4.1 (1.8–9.4)	0.001	2.3 (0.9–5.9)	0.072
	**OR (95% CI)**	** *p* **	**OR (95% CI)**	** *p* **
Radiotherapy	5.3 (1.8–15.1)	0.002	0.6 (0.1–3.9)	0.625
Chemotherapy	1.2 (0.4–3.8)	0.727	NI	NI
Surgery	2.7 (0.9–8.3)	0.072	NI	NI
Number of concomitant APD	1.8 (1.1 -3)	0.014	2 (1.1–3.8)	0.034
Number of progressions	1.2 (0.9–1.6)	0.095	NI	
Years of follow-up	1.2 (1.1–1.3)	0.001	1 (0.8–1.2)	0.888

MD: metabolic disorder; MDC: modified Dodge classification, HR: hazard ratio; OR: Odds Ratio, CI: confidence interval. NI: not included (since not statistically significant in univariate model).

## Data Availability

The data presented in this study are available on request from the corresponding author.

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
