# Peer review of "A 40-Year Cohort Study of Evolving Hypothalamic Dysfunction in Infants and Young Children (<3 years) with Optic Pathway Gliomas"

_cancers, 2022, doi:10.3390/cancers14030747_

Round 1
Reviewer 1 Report
Picariello and Cerbone et al. have submitted an informative review of 90 children under the age of 3 years with optic pathway gliomas and subsequent hypothalamic dysfunction. This was a 40-year cohort study with a median follow-up time of 9.5 years. The long term complications of a low grade optic glioma were assessed, including hormonal, metabolic and neuro-behavioural disorders. Treatment factors were related to outcome. The manuscripts gives a clear overview of development of endocrine disorders during follow-up and the sequence of events.
Minor comments
- Title suggests that children with hypothalamic dysfunction are included, whilst only n = 56 had hypothalamic involvement.
- I think the introduction deserves some extra background information such as; what is currently known about the evolution of hypothalamic dysfunction in these young children?
- Line 93 and line 104: write down numbers and percentages consistent throughout the paper; so for example 27: 30% instead of 27/90:30%
- Line 94: what was the mean age of presentation in children with NF 1 tumors?]
- What were the specific causes of death?
- Figure 1A: because of the low numbers of patients that were followed for more than 15 years it gives a distorted view of the overall survival which now seems lower than expected. Would be informative if numbers could be included below the figure.
- Line 123: what was the median age at radiotherapy treatment?
- Line 195: according to the range, in girls, CPP treatment in some was only started at the age of 8.5 years. Were these children diagnosed < 8 years?
- Were the curves adjusted for competing risks? For example; you cannot get anterior pituitary deficiency if you have already died.
- Line 221: odds ratio of surgical intervention is not in line with the odds ratio included in the table above.
- Line: 328 include reference
- Multivariable analysis: how was the decision for cox or logistic regression made? Is it true that multivariable did not include all 10 factors but only 5 of them (since these are 2 distinct analysis methods?) If that is the case it would mean that for example hypothalamic involvement was not corrected for radiotherapy?
- Figures supplement: Changing the curves to cumulative incidence curves instead of survival curves would give a clearer view.
Major comments
- Line 82: our search identified. How were patients searched? Could this have led to any bias?
- Definitions:
- What is considered surgery? Is hydrocephalus surgery included? Or VP-drain?
- How was hypothalamic involvement scored? By a radiologist? Which scoring system?
- How was raised intracranial pressure defined and scored?
- What was the definition of neuro-behavioural problems?
- How many children presented with endo-metabolic dysfunction or developed this in the 1st 6 months after diagnosis? ? How many had CPP at diagnosis? GH hypersecretion or DI? Was the presence of HP dysfunction at diagnosis related to endo-metabolic dysfunction at follow-up?
- Regarding the multivariable analysis: more than one factor per 10 outcomes were included. For example table 4, MD. Only 23 children developed MD but 10 factors were included in this analysis, making it less reliable. Please adjust this analysis.
Reviewer 2 Report
The authors present the long-term endocrine, metabolic, and educational status of a group of 90 children diagnosed with optic pathway-hypothalamic gliomas before 3 years of age. The major strength of the manuscript is the long-term outcome with the median of over 10 years; this type of perspective is sorely lacking in the literature.
There are several areas that I believe should be addressed to further strengthen the manuscript:
1. Optic pathway-hypothalamic glioma is presenting early in childhood have been noted to frequently be pilomyxoid astrocytomas, a more aggressive variant than the typical pilocytic astrocytoma seen in older children. It would be worthwhile to correlate the pathology if available with the outcome
2. Unlike older children with optic pathway-hypothalamic glioma's, infants and younger children will often present with massive lesions. Going beyond the modified Dodge criteria, it would be useful to know if size or volume of tumor at presentation had an impact.
3. Although the authors clearly state that both radiotherapy and surgery have unique complications, both short-term and long-term (e.g. moyamoya with radiation and posterior pituitary dysfunction with surgery), compared to chemotherapy it is unclear whether the response to chemotherapy with tumor shrinkage has any beneficial impact on the long-term endocrine, metabolic, and educational outcomes.
4. It is unclear how the presence of a shunt procedure increased morbidity. Is it in epiphenomena associated with massive tumors rather than the procedure itself?
Overall an excellent manuscript that definitely should be published
Author Response
POINT BY POINT REPLY
We thank the reviewers for their useful comments.
Point by point replies are listed below and the relevant changes are highlighted in yellow in the manuscript.
Reviewer 2
The authors present the long-term endocrine, metabolic, and educational status of a group of 90 children diagnosed with optic pathway-hypothalamic gliomas before 3 years of age. The major strength of the manuscript is the long-term outcome with the median of over 10 years; this type of perspective is sorely lacking in the literature.
There are several areas that I believe should be addressed to further strengthen the manuscript:
- Optic pathway-hypothalamic glioma is presenting early in childhood have been noted to frequently be pilomyxoid astrocytomas, a more aggressive variant than the typical pilocytic astrocytoma seen in older children. It would be worthwhile to correlate the pathology if available with the outcome
Thank you for your comment. We agree, it would be worth to correlate pathology with the outcome, but unfortunately pathology was only available for half of children (48/90: 53%) since it is a retrospective study and biopsy was not systematically performed in all children, especially in the past. However, to provide the data about the pathology results in those cases who underwent surgery, we included the following sentence in the result section at lines 113-115: “Pathology was available in 48 children who underwent surgery: 39 tumours were Pilocytic Astrocytoma, 5 Pilomyxoid Astrocytoma, 3 Ganglioglioma and 1 Fibrillary Astrocytoma.”
- Unlike older children with optic pathway-hypothalamic glioma's, infants and younger children will often present with massive lesions. Going beyond the modified Dodge criteria, it would be useful to know if size or volume of tumor at presentation had an impact.
Thank you for raising this point. Unfortunately, since this is a retrospective study, T2 weighted volumetric sequences were not available for all patients.
Regarding 2D tumour size, RANO criteria are utilized to determine the percentage of change and tumour response within the same patients over time. However, as previously reported in a study published by our institution and The Hospital for Sick Children Toronto (Volumetric assessment of tumor size changes in pediatric low-grade gliomas: feasibility and comparison with linear measurements” D'Arco F, O'Hare P, Dashti F, Lassaletta A, Loka T, Tabori U, Talenti G, Thust S, Messalli G, Hales P, Bouffet E, Laughlin S. Neuroradiology. 2018 Apr;60(4):427-436), the assessment according to RANO criteria is not reliable for complex shaped lesions such as paediatric optic gliomas, that often have mixed solid and cystic components. Moreover, the radiologist decides the slice that should correspond to the largest appearing cross-section image of the tumour and slight variations on serial imaging might affect reproducibility of measurements in irregular lesions. This might lead to incorrect size estimation in several cases. In fact, bi-dimensional measurements differed from volumetric results in 20% of cases in our previously published study. In view of the above limitations, we decided not to perform and collect 2D measurements for evaluation of the tumour size in order to avoid bias and unreliable results.
We agree that this is a very interesting and relevant topic and should be addressed in a dedicated study
- Although the authors clearly state that both radiotherapy and surgery have unique complications, both short-term and long-term (e.g. moyamoya with radiation and posterior pituitary dysfunction with surgery), compared to chemotherapy it is unclear whether the response to chemotherapy with tumor shrinkage has any beneficial impact on the long-term endocrine, metabolic, and educational outcomes.
As highlighted by univariate logistic regression results, it seems that chemotherapy increases the risk of anterior pituitary disorders (OR 4, 95% CI 1.4 – 11.6, p 0.009), whilst it has no impact on metabolic (OR 1.2, 95% CI 0.4 – 3.8, p 0.727) and neuro-behavioural outcomes (OR 1.2, 95% CI 0.4 – 3.1, p 0.724). The latter results are not reported in the manuscript since we only mentioned the significant variables for neuro-behavioural outcomes.
In detail, regarding the effect of chemotherapy on anterior pituitary disorders, we believe that it might be related to a more aggressive behaviour of larger tumours, especially suprasellar masses involving the hypothalamus, with failure of chemotherapy to stabilise them and therefore undergoing multiple progressions and treatment strategies, more than to the effect of chemotherapy itself. This hypothesis is supported by the fact that chemotherapy is not significant in multivariable binary logistic regression model.
The impact of tumour stability or shrinkage on any evolving neuroendocrine dysfunction is important to ascertain but difficult to achieve long-term, especially when individualised sequential treatment interventions are applied to an age-dependent biphasic evolutionary disease course. This would require future trials designed with this important outcome in mind, targeting this specific, high-risk age-cohort for early neuroendocrine and neurodevelopmental support, followed carefully throughout the age span.
In our population, 41 children with chiasmatic/post-chiasmatic tumours were treated with chemotherapy as initial treatment strategy, without concomitant surgery (decompressive surgery, biopsy or resection of any extent). The majority of these children (85%, 35/41) experienced disease progression and were subjected to repeated treatments, whilst only 6 children never progressed. Five out of 6 patients, never developed endo-metabolic disorders (3 had hypothalamic involvement), whilst a subject with a MDC3 H+ tumour developed GHD.
However, it is impossible to draw general conclusions on such small numbers.
In view of the above data, the hypothesis of performing an EEFS analysis with patient stratification according to chemotherapy response (tumor shrinkage of more than 25%) seems unfeasible as it would generate unreliable results.
In fact, in order to avoid confounding factors, patients without events should be censored prior to progression or initiation of further treatment. This might lead to a limited follow-up time and would provide short term rather than long-term results.
We believe that the impact of chemotherapy and tumour shrinkage on the long-term morbidity is very difficult to explore in such a young population with prolonged disease course and multiple treatments, and the answer still remains an open issue.
The above appoints have been summarised at lines 420-425.
- It is unclear how the presence of a shunt procedure increased morbidity. Is it in epiphenomena associated with massive tumors rather than the procedure itself?
Yes, we do believe that this is likely to be an epiphenomena associated with bigger tumour.
Most patients undergoing decompressive surgery also underwent resections (with only 6 patients receiving decompressive surgery only), hence discriminating the effect of the two different types of surgeries in a study retrospective in nature and in patients undergoing multiple and repeated treatments, would be rather difficult.
Overall an excellent manuscript that definitely should be published